# Two-dimensional video-based analysis of human gait using pose estimation

**Jan Stenum** [1,2], **Cristina Rossi**[1,3], **Ryan T. Roemmich**[1,2] *

**1** Center for Movement Studies, Kennedy Krieger Institute, Baltimore, Maryland, United States of America, **2** Department of Physical Medicine and Rehabilitation, Johns Hopkins University School of Medicine, Baltimore, Maryland, United States of America, **3** Department of Biomedical Engineering, Johns Hopkins University, Baltimore, Maryland, United States of America

* rroemmi1@jhmi.edu

**Data Availability Statement:** All data files are available from http://bytom.pja.edu.pl/projekty/hm-gpjatk/. Software is available from https://github. com/janstenum/GaitAnalysis-PoseEstimation.

## Abstract

Human gait analysis is often conducted in clinical and basic research, but many common approaches (e.g., three-dimensional motion capture, wearables) are expensive, immobile, data-limited, and require expertise. Recent advances in video-based pose estimation suggest potential for gait analysis using two-dimensional video collected from readily accessible devices (e.g., smartphones). To date, several studies have extracted features of human gait using markerless pose estimation. However, we currently lack evaluation of video-based approaches using a dataset of human gait for a wide range of gait parameters on a stride-by-stride basis and a workflow for performing gait analysis from video. Here, we compared spatiotemporal and sagittal kinematic gait parameters measured with OpenPose (open-source video-based human pose estimation) against simultaneously recorded three-dimensional motion capture from overground walking of healthy adults. When assessing all individual steps in the walking bouts, we observed mean absolute errors between motion capture and OpenPose of 0.02 s for temporal gait parameters (i.e., step time, stance time, swing time and double support time) and 0.049 m for step lengths. Accuracy improved when spatiotemporal gait parameters were calculated as individual participant mean values: mean absolute error was 0.01 s for temporal gait parameters and 0.018 m for step lengths. The greatest difference in gait speed between motion capture and OpenPose was less than 0.10 m s$^{-1}$. Mean absolute error of sagittal plane hip, knee and ankle angles between motion capture and OpenPose were 4.0˚, 5.6˚ and 7.4˚. Our analysis workflow is freely available, involves minimal user input, and does not require prior gait analysis expertise. Finally, we offer suggestions and considerations for future applications of pose estimation for human gait analysis.

## Author summary

There is a growing interest among clinicians and researchers to use novel pose estimation algorithms that automatically track human movement to analyze human gait. Gait analysis is routinely conducted in designated laboratories with specialized equipment. On the

**Funding:** This study was funded by NIH grant R21AG059184 to RTR. The funder had no role in study design, data collection and analysis, decision to publish, or preparation of the manuscript.

**Competing interests:** The authors have declared that no competing interests exist.

other hand, pose estimation relies on digital videos that can be recorded from household devices such as a smartphone. As a result, these new techniques make it possible to move beyond the laboratory and perform gait analysis in other settings such as the home or clinic. Before such techniques are adopted, we identify a critical need for comparing outcome parameters against three-dimensional motion capture and to evaluate how camera viewpoint affect outcome parameters. We used simultaneous motion capture and left- and right-side video recordings of healthy human gait and calculated spatiotemporal gait parameters and lower-limb joint angles. We find that our provided workflow estimates spatiotemporal gait parameters together with hip and knee angles with the accuracy and precision needed to detect changes in the gait pattern. We demonstrate that the position of the participant relative to the camera affect spatial measures such as step length and discuss the limitations posed by the current approach.

## Introduction

Humans have been interested in studying the walking patterns of animals and other humans for centuries, dating back to Aristotle in the fourth century BC (see Baker [1] for a detailed history of gait analysis). Gait analysis technologies have evolved from Borelli's use of staggered poles to study his own gait to modern tools that include three-dimensional motion capture, instrumented gait mats, and a variety of wearable devices. Although technological advances continue to expand our abilities to measure human walking in clinical and laboratory settings, many limitations persist. Current techniques remain expensive, are often time consuming, and require specialized equipment or expertise that is often not widely accessible.

Recent progress in video-based pose estimation has enabled automated analysis of the movements of humans [2–7] and animals [8,9] using only digital video inputs. The learning algorithms at the core of human pose estimation approaches use networks that are generally trained on many images of different people (e.g., MPII [2] and COCO [10] datasets), resulting in robust networks capable of detecting keypoints (e.g., body landmarks) in new images beyond the training dataset. These software packages are freely available and have the potential to expand the ability to generate large datasets of human gait data by enabling data collection in any setting (including the home or clinic) with little cost of time, money, or effort.

Several prior studies have extracted features of human gait using markerless pose estimation [11–17]; however, there remains a critical need for comparisons of these techniques against simultaneously collected, gold-standard measurements. We regard the following considerations as imperative to evaluate the performance of markerless pose estimation for human gait analysis: 1) use of a dataset containing overground walking sequences with synchronized three-dimensional motion capture and video recordings, including video recordings from multiple perspectives, 2) stride-by-stride comparisons of gait parameters in addition to comparisons of average gait parameters across walking bouts for individual participants, 3) comparisons of a wide range of gait parameters including spatiotemporal and kinematic measures and 4) an approach with no need for additional network processing. In addition, there is a need for dissemination of a workflow that produces gait parameter outputs from a simple digital video input that is quick, involves minimal user input and does not require prior gait analysis expertise.

The goals of this study were two-fold: 1) compare spatiotemporal and kinematic gait parameters as measured by simultaneous recordings of three-dimensional motion capture and pose estimation via OpenPose, a freely available human pose estimation algorithm that uses

Part Affinity Fields to detect up to 135 keypoints (using models of "body", "foot", "hand" and "facial" keypoints) in images of humans [3,6], and 2) provide a workflow and suggestions for performing automated human gait analysis from digital video. First, we used OpenPose to detect keypoints in videos of healthy adults walking overground. These videos were provided in a freely available dataset that includes synchronized digital videos and three-dimensional motion capture gait data [18]. We then developed a workflow to calculate a variety of spatio-temporal and kinematic gait parameters from the OpenPose outputs. We compared these parameters against measurements calculated from the motion capture data and across different camera views within the same walking trials to test the robustness of using OpenPose to estimate gait parameters from different camera viewpoints.

We wish to state upfront that the workflow provided here is simply one approach to human gait analysis using pose estimation. We tested OpenPose by using the available demo without modification, as we anticipate that this is the most accessible way for new users with little expertise in computer science or engineering to run the software. We consider that there may be opportunity to improve upon the results presented in the current study by adjusting Open-Pose parameters, using other pose estimation algorithms, or using other methods of video recording. We do not claim to provide an optimized approach to human gait analysis through video-based pose estimation, but rather an approach that we found to be easy to use and fast.

## Results

We used a dataset [18] containing three-dimensional motion capture and sagittal plane video recordings from left and right viewpoints of overground walking of healthy adults (Fig 1). Video recordings were analyzed with OpenPose [3] and subsequent post-processing in MATLAB (post-processing scripts made freely available at https://github.com/janstenum/GaitAnalysis-PoseEstimation). Timings of gait events (heel-strikes and toe-offs), spatiotemporal gait parameters (step time, stance time, swing time, double support time, step length and gait speed) and lower extremity sagittal joint angles of the hip, knee and ankle were independently calculated for motion capture and for left and right viewpoints of OpenPose.

### Event times

First, we examined how well OpenPose identified common gait events (i.e., heel-strikes and toe-offs) compared to event times identified in motion capture data (Table 1, leftmost and middle columns in each section). The group mean difference (a measure of bias between measurement systems) in heel-strike detection between motion capture and OpenPose left ($C_L$) or right ($C_R$) side views was up to one motion capture frame (10 ms; sampling frequency of motion capture data was 100 Hz), the group mean *absolute* difference (a measure of the error between measurement systems) was up to two motion capture frames (20 ms) and the greatest difference for any individual heel-strike detection was six motion capture frames (60 ms). Group mean difference in toe-off detection between motion capture and OpenPose left- or right-side views was up to two motion capture frames (20 ms), mean absolute difference was up to three motion capture frames (30 ms) and the greatest difference for any individual toe-off detection was 11 motion capture frames (110 ms).

Next, we evaluated how event times compared across OpenPose left and right viewpoints (Table 1, rightmost column in each section). The mean difference and absolute difference in heel-strike and toe-off events was less than one video frame (40 ms; sampling frequency of video recordings was 25 Hz); the greatest difference was two (80 ms) and three (120 ms) video frames for heel-strikes and toe-off events, respectively.

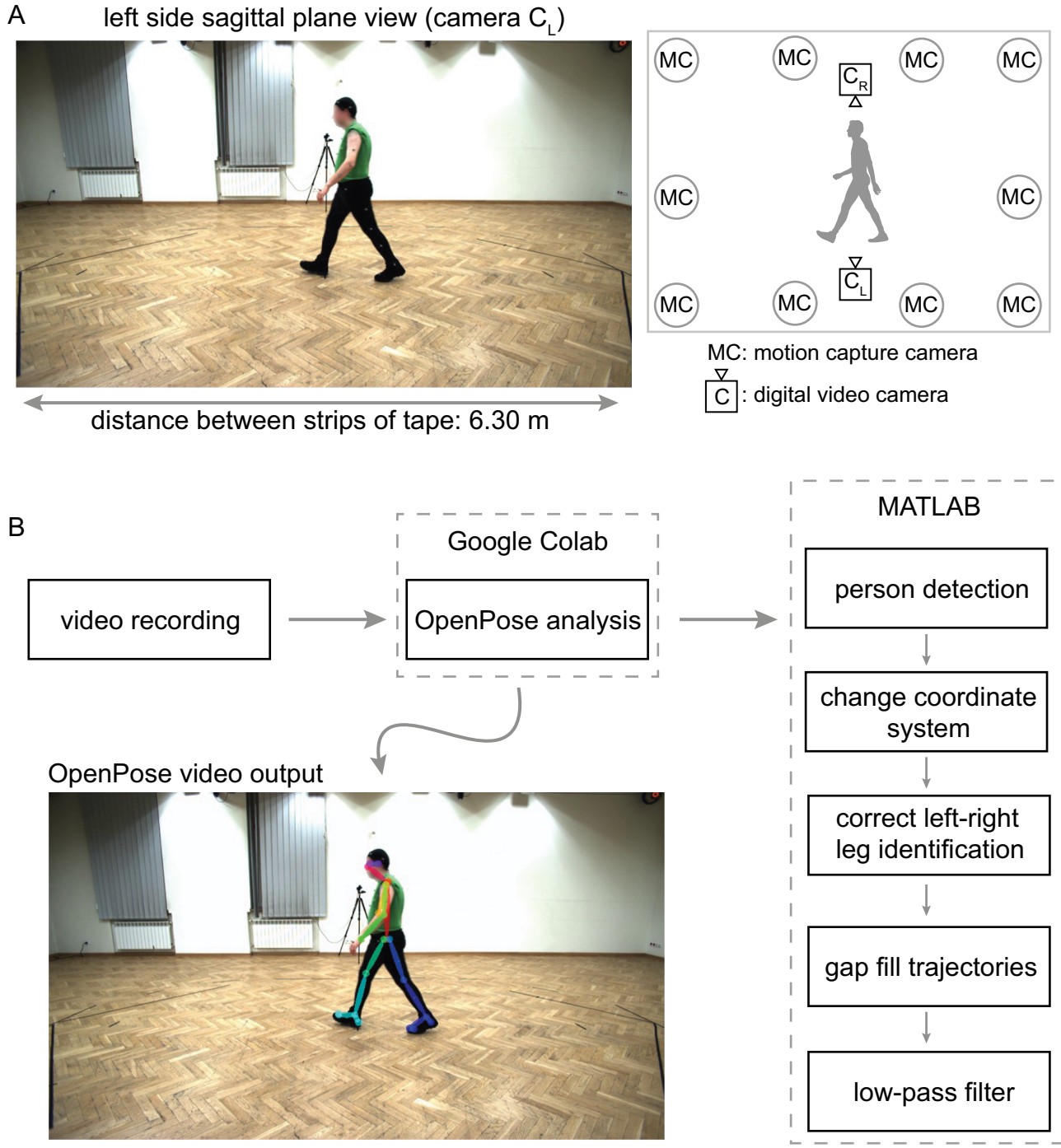

**Fig 1. Overview of laboratory space and workflow.** A) Representative image frame of original video recording from left-side sagittal plane view with diagram of motion capture and video cameras. We used a known distance of 6.30 m between two strips of tape on the floor to dimensionalize pixel coordinates of OpenPose keypoints. The public dataset [18] that we used is made available at http://bytom.pja.edu.pl/projekty/hm-gpjatk/. See Release Agreement for copyrights and permissions. B) Workflow of video recordings (available at https://github.com/janstenum/GaitAnalysis-PoseEstimation). We analyzed video recordings with OpenPose using Google Colaboratory and next processed the data using custom MATLAB scripts.

**Table 1. Differences in event times for all steps.** *MC*: motion capture; $C_L$: OpenPose left-side view; $C_R$: OpenPose right-side view. Asterisks (*) denote $P < 0.05$.

| | N | mean±SD | | | mean±SD | | | range | | |
|---|---|---|---|---|---|---|---|---|---|---|
| | | $MC-C_L$ | $MC-C_R$ | $C_L-C_R$ | $\|MC-C_L\|$ | $\|MC-C_R\|$ | $\|C_L-C_R\|$ | $MC-C_L$ | $MC-C_R$ | $C_L-C_R$ |
| Left Heel-Strike Time (s) | 107 | −0.01±0.02 | 0.00±0.02 | 0.00±0.02 | 0.02±0.01 | 0.01±0.01 | 0.01±0.02 | [−0.05, 0.06] | [−0.05, 0.04] | [−0.08, 0.04] |
| Right Heel-Strike Time (s) | 109 | −0.01±0.02 | −0.01±0.02 | 0.00±0.02 | 0.01±0.01 | 0.01±0.01 | 0.01±0.02 | [−0.04, 0.04] | [−0.05, 0.03] | [−0.04, 0.04] |
| Left Toe-Off Time (s) | 109 | −0.01±0.03 | −0.01±0.02 | 0.00±0.04 | 0.03±0.02 | 0.02±0.02 | 0.02±0.03 | [−0.09, 0.06] | [−0.08, 0.04] | [−0.12, 0.08] |
| Right Toe-Off Time (s) | 107 | −0.02±0.02 | −0.01±0.03 | 0.01±0.03 | 0.02±0.02 | 0.02±0.02 | 0.02±0.02 | [−0.11, 0.05] | [−0.07, 0.07] | [−0.04, 0.12] |

## Spatiotemporal gait parameters

**Temporal parameters–all steps.** The group mean difference in temporal gait parameters (step time, stance time, swing time and double support time; compared for all individual steps in the walking bouts) between motion capture and OpenPose left- or right-side views was up to one motion capture frame (10 ms), the mean absolute difference in temporal gait parameters was two motion capture frames (20 ms) and the greatest difference was 10 motion capture frames (100 ms) (Table 2). Post-hoc tests indicated statistical differences between motion capture and OpenPose for stance time, swing time and double support (Table 3). Pearson and intra-class correlation coefficients between motion capture and OpenPose were strong for step time, stance time and swing time (Fig 2A–2C and Table 3, coefficients between 0.839 and 0.971), but less strong for double support time (Fig 2D and Table 3, coefficients between 0.660 and 0.735).

When comparing OpenPose left- and right-side views, the group mean difference and absolute difference in temporal gait parameters was less than one video frame (40 ms) and the greatest difference was three video frames (120 ms) (Table 2). Temporal gait parameters were not statistically different between OpenPose left and right views (Table 3). Pearson and intra-class correlation coefficients between OpenPose views were strong for step time, stance time and swing time (Fig 2A–2C and Table 3, coefficients between 0.809 and 0.951), but less strong for double support time (Fig 2D and Table 3, coefficients between 0.522 and 0.524).

**Temporal parameters–individual participant means.** Group mean differences and absolute differences in temporal gait parameters (compared for individual participant mean values across the walking bout) between motion capture and OpenPose left- and right-side views were up to one motion capture frame (10 ms) and the greatest difference was four motion capture frames (40 ms) (Table 4). Post-hoc tests indicated statistically significant differences between motion capture and OpenPose for stance time, swing time and double support time (Table 5). Pearson and intra-class correlation coefficients between motion capture and

**Table 2. Gait parameters calculated for all steps.** *MC*: motion capture; $C_L$: OpenPose left-side view; $C_R$: OpenPose right-side view.

| | N | mean±SD | | | mean±SD | | | mean±SD | | | range | | |
|---|---|---|---|---|---|---|---|---|---|---|---|---|---|
| | | MC | $C_L$ | $C_R$ | $MC-C_L$ | $MC-C_R$ | $C_L-C_R$ | $\|MC-C_L\|$ | $\|MC-C_R\|$ | $\|C_L-C_R\|$ | $MC-C_L$ | $MC-C_R$ | $C_L-C_R$ |
| Step Time (s) | 185 | 0.61 ±0.08 | 0.61 ±0.08 | 0.61 ±0.08 | 0.00±0.02 | 0.00 ±0.02 | 0.00 ±0.03 | 0.02 ±0.01 | 0.02 ±0.01 | 0.02 ±0.03 | [−0.06, 0.08] | [−0.06, 0.06] | [−0.12, 0.08] |
| Stance Time (s) | 154 | 0.75 ±0.10 | 0.75 ±0.11 | 0.75 ±0.10 | −0.01 ±0.03 | −0.01 ±0.02 | 0.00 ±0.03 | 0.02 ±0.02 | 0.02 ±0.02 | 0.02 ±0.02 | [−0.10, 0.06] | [−0.06, 0.05] | [−0.08, 0.12] |
| Swing Time (s) | 154 | 0.46 ±0.04 | 0.45 ±0.05 | 0.45 ±0.05 | 0.01±0.03 | 0.01 ±0.02 | 0.00 ±0.03 | 0.02 ±0.02 | 0.02 ±0.02 | 0.02 ±0.02 | [−0.06, 0.07] | [−0.05, 0.07] | [−0.08, 0.08] |
| Double Support Time (s) | 154 | 0.14 ±0.03 | 0.15 ±0.04 | 0.15 ±0.04 | −0.01 ±0.03 | −0.01 ±0.03 | 0.00 ±0.04 | 0.02 ±0.02 | 0.02 ±0.02 | 0.02 ±0.03 | [−0.08, 0.05] | [−0.07, 0.07] | [−0.08, 0.08] |
| Step Length (m) | 215 | 0.598 ±0.095 | 0.606 ±0.118 | 0.598 ±0.114 | −0.008 ±0.059 | 0.000 ±0.059 | 0.008 ±0.094 | 0.049 ±0.033 | 0.047 ±0.035 | 0.076 ±0.056 | [−0.106, 0.202] | [−0.112, 0.204] | [−0.282, 0.253] |

**Table 3. Statistical tests for gait parameters calculated for all steps.** *MC*: motion capture; $C_L$: OpenPose left-side view; $C_R$: OpenPose right-side view. Asterisks (*) denote $P < 0.05$.

| | *F* | post-hoc (*P*) | | | *r* | | | ICC$_{C-1}$ | | | ICC$_{A-1}$ | | |
|---|---|---|---|---|---|---|---|---|---|---|---|---|---|
| | | *MC v $C_L$* | *MC v $C_R$* | *$C_L$ v $C_R$* | *MC v $C_L$* | *MC v $C_R$* | *$C_L$ v $C_R$* | *MC v $C_L$* | *MC v $C_R$* | *$C_L$ v $C_R$* | *MC v $C_L$* | *MC v $C_R$* | *$C_L$ v $C_R$* |
| Step Time | 0.2 | - | - | - | 0.957* | 0.963* | 0.911* | 0.955* | 0.962* | 0.911* | 0.955* | 0.963* | 0.911* |
| Stance Time | 9.7* | <0.001 | <0.001 | 1.000 | 0.967* | 0.971* | 0.951* | 0.966* | 0.971* | 0.950* | 0.962* | 0.967* | 0.950* |
| Swing Time | 12.3* | <0.001 | <0.001 | 1.000 | 0.860* | 0.858* | 0.810* | 0.855* | 0.857* | 0.809* | 0.839* | 0.842* | 0.810* |
| Double Support Time | 8.0* | 0.001 | <0.001 | 1.000 | 0.691* | 0.735* | 0.523* | 0.678* | 0.714* | 0.522* | 0.660* | 0.694* | 0.524* |
| Step Length | 1.6 | - | - | - | 0.869* | 0.857* | 0.671* | 0.849* | 0.843* | 0.671* | 0.848* | 0.844* | 0.671* |

OpenPose were strong (Fig 3 and Table 5, coefficients between 0.861 and 0.998) for all temporal gait parameters.

When comparing OpenPose left- and right-side views, the group mean difference, absolute difference and the greatest difference were all less than one video frame (40 ms) (Table 4). Temporal gait parameters were not statistically different between OpenPose left and right views (Table 5). Pearson and intra-class correlation coefficients between OpenPose views were strong (Fig 3 and Table 5, coefficients between 0.893 and 0.995) for all temporal gait parameters.

**Step length and gait speed.** When step lengths were calculated on a step-by-step basis (Fig 4A and Table 2), the group mean difference between motion capture and OpenPose left- or right-side views was less than 0.010 m, mean absolute difference was 0.049 m and the greatest difference was 0.204 m. The group mean difference between OpenPose left and right views was less than 0.010 m, mean absolute difference was 0.076 m and the greatest difference was 0.282 m.

The agreement in step length calculations between measurement systems improved when step length was calculated as individual participant means instead of step-by-step (Fig 4B and Table 4): group mean difference between motion capture and OpenPose was less than 0.010 m, mean absolute difference was less than 0.020 m and the greatest difference was 0.050 m; group mean difference between OpenPose left- and right-side views was less than 0.010 m, mean absolute difference was 0.020 m and the greatest difference was 0.057 m. Pearson and intra-class correlation coefficients also suggest better estimation of step lengths when calculated as individual participant means: correlation coefficients were higher when step lengths were calculated as individual participant means (Fig 4B and Table 5, coefficients between 0.927 and 0.973) compared to when calculated for all steps (Fig 4A and Table 3, coefficients between 0.671 and 0.869).

When examining individual step length data, we occasionally observed large discrepancies between measurement systems. For example, the maximal difference of step length between motion capture and OpenPose was 0.204 m, which was substantial given that the length of this step was measured to be 0.672 m by motion capture (i.e., the discrepancy was greater than 30% of the step length). We surmised that anterior-posterior position of the participant on the walkway could have affected how well OpenPose estimated step lengths because parallax and changes in perspective could have affected the video-based analyses. We therefore performed a secondary analysis to investigate how anterior-posterior position of the participant along the walkway affected differences in step lengths between the measurement systems.

When step length differences are plotted against anterior-posterior position of the C7 marker (Fig 5), it is apparent that the anterior-posterior position of the participant along the walkway affects the step length estimate using OpenPose. When step length is calculated at left heel-strike from a left ($C_L$) side view (Fig 5A, dark circles), step length differences are positive

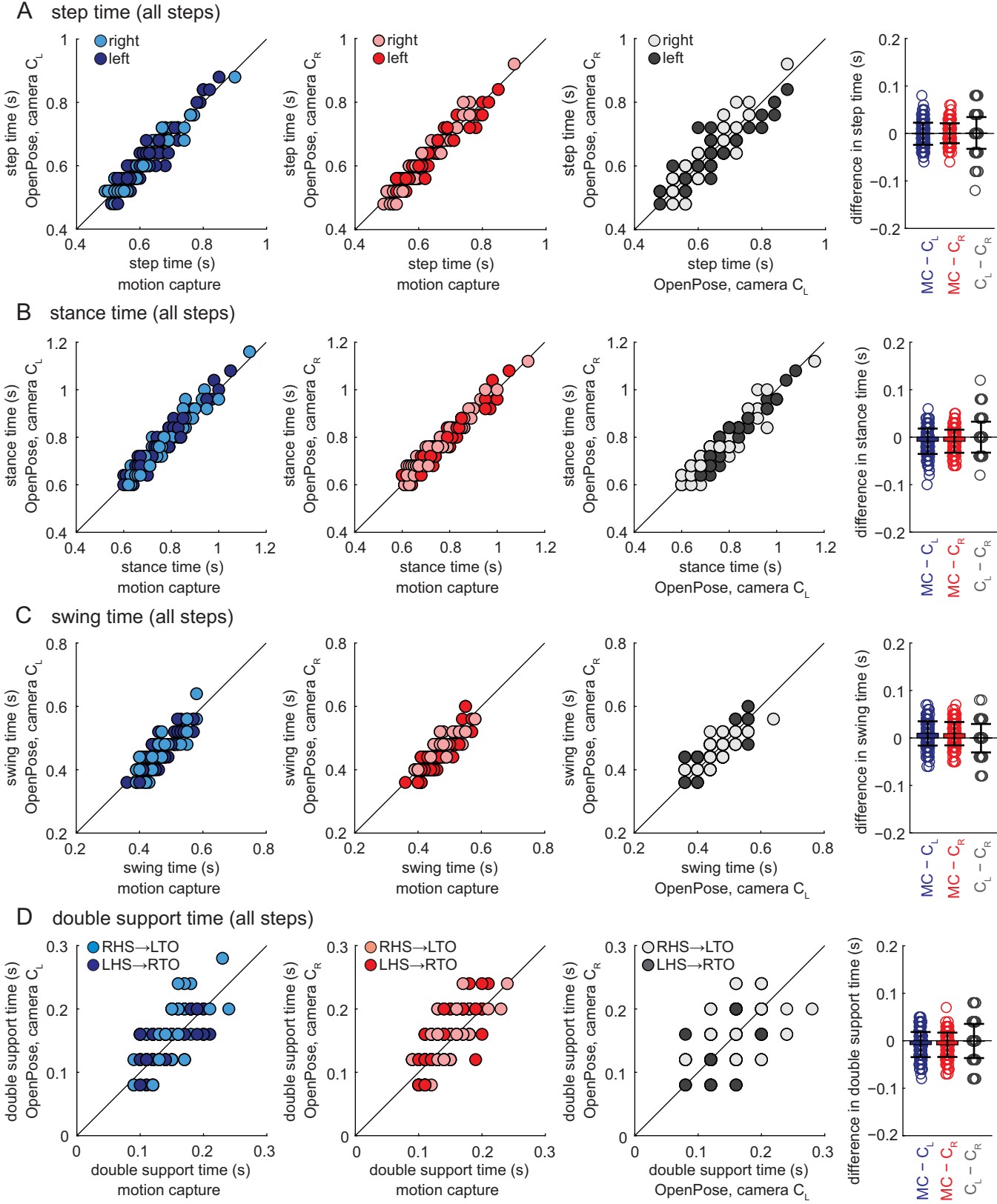

**Fig 2. Temporal gait parameters for all individual steps shown for all participants and measurement systems.** Panels show A) step time, B) stance time, C) swing time, and D) double support time. Data shown in blue represent comparisons between motion capture and OpenPose left-side ($C_L$) views, data shown in red represent comparisons between motion capture and OpenPose right-side ($C_R$) views, and data shown in gray represent comparisons between the two OpenPose views ($C_L$ and $C_R$). Dark circles represent left leg data, light circles represent right leg data. Color schemes and shading are consistent across all similar figures (Figs 3 and 4). Bar plots on the far right show individual data, group means, and SD to visualize the distribution of the differences observed between the measurement systems. Results from statistical analyses are shown in Tables 2 and 3.

(motion capture greater than OpenPose) at the start of the walkway and differences gradually become negative (motion capture less than OpenPose) as the participant traverses the walkway. Differences in step lengths at right heel-strike between motion capture and OpenPose left ($C_L$) side view show the opposite trend (Fig 5A, light circles): differences are negative (motion capture less than OpenPose) at the start of the walkway and gradually become positive (motion capture greater than OpenPose) at the end of the walkway. Note that differences in step length at left and right heel-strikes overlap and are minimized at the middle of the walkway when the person is in the center of the field of view of the cameras. These effects can be observed in the representative images shown in Fig 6.

Group mean difference in gait speed (Fig 4C and Table 4, calculated as the average speed of the walking trial) between measurement systems was up to 0.01 m s$^{-1}$, mean absolute difference was up to 0.04 m s$^{-1}$ and the greatest difference between measurement systems was 0.09 m s$^{-1}$. Gait speed was not statistically different between measurement systems and Pearson and intra-class correlation coefficients were strong (Table 5, coefficients between 0.964 and 0.986).

## Sagittal lower extremity joint kinematics

Next, we examined how well OpenPose estimated sagittal lower extremity joint angles. We calculated sagittal plane hip, knee and ankle angles across the stride cycle that were averaged for the walking bout of each individual participant (Fig 7A). Mean absolute error of joint angles between motion capture and OpenPose left- or right-side views were 4.0° for the hip, 5.6° for the knee and 7.4° for the ankle (Table 6). This suggests that ankle angles were the least accurate of the three joints when comparing OpenPose to motion capture data. This is supported by cross-correlation coefficients (measured at time lag zero) showing that the time-varying nature of joint angles across the gait cycle is better preserved across comparisons between motion capture and OpenPose for the hip and knee (Fig 7B and Table 7, coefficients between 0.972 and 0.984) than the ankle (coefficients between 0.743 and 0.778). Mean absolute error between

**Table 4. Gait parameters calculated as individual participant means.** $MC$: motion capture; $C_L$: OpenPose left-side view; $C_R$: OpenPose right-side view.

| | N | mean±SD | | | mean±SD | | | mean±SD | | | range | | |
|---|---|---|---|---|---|---|---|---|---|---|---|---|---|
| | | $MC$ | $C_L$ | $C_R$ | $MC-C_L$ | $MC-C_R$ | $C_L-C_R$ | $\lvert MC-C_L\rvert$ | $\lvert MC-C_R\rvert$ | $\lvert C_L-C_R\rvert$ | $MC-C_L$ | $MC-C_R$ | $C_L-C_R$ |
| Step Time (s) | 31 | 0.61 ±0.07 | 0.61 ±0.07 | 0.61 ±0.07 | 0.00±0.00 | 0.00 ±0.00 | 0.00 ±0.01 | 0.00 ±0.00 | 0.00 ±0.00 | 0.00 ±0.00 | [−0.01, 0.01] | [−0.01, 0.01] | [−0.02, 0.01] |
| Stance Time (s) | 31 | 0.74 ±0.09 | 0.75 ±0.09 | 0.75 ±0.09 | −0.01 ±0.01 | −0.01 ±0.01 | 0.00 ±0.01 | 0.01 ±0.01 | 0.01 ±0.01 | 0.01 ±0.01 | [−0.04, 0.01] | [−0.03, 0.01] | [−0.02, 0.02] |
| Swing Time (s) | 31 | 0.46 ±0.04 | 0.45 ±0.04 | 0.45 ±0.04 | 0.01±0.01 | 0.01 ±0.01 | 0.00 ±0.01 | 0.01 ±0.01 | 0.01 ±0.01 | 0.01 ±0.01 | [−0.01, 0.03] | [−0.02, 0.03] | [−0.02, 0.02] |
| Double Support Time (s) | 31 | 0.14 ±0.03 | 0.15 ±0.03 | 0.15 ±0.03 | −0.01 ±0.01 | −0.01 ±0.01 | 0.00 ±0.01 | 0.01 ±0.01 | 0.01 ±0.01 | 0.01 ±0.01 | [−0.03, 0.01] | [−0.03, 0.02] | [−0.03, 0.02] |
| Step Length (m) | 31 | 0.603 ±0.060 | 0.611 ±0.063 | 0.603 ±0.065 | −0.008 ±0.020 | 0.000 ±0.015 | 0.008 ±0.023 | 0.018 ±0.012 | 0.011 ±0.010 | 0.020 ±0.014 | [−0.050, 0.031] | [−0.030, 0.038] | [−0.052, 0.057] |
| Gait Speed (m s$^{-1}$) | 31 | 1.00 ±0.15 | 1.02 ±0.16 | 1.01 ±0.16 | −0.01 ±0.04 | 0.00 ±0.03 | 0.01 ±0.04 | 0.03 ±0.02 | 0.02 ±0.02 | 0.04 ±0.02 | [−0.09, 0.05] | [−0.06, 0.07] | [−0.09, 0.08] |

**Table 5. Statistical tests for gait parameters calculated as individual participant means.** *MC*: motion capture; $C_L$: OpenPose left-side view; $C_R$: OpenPose right-side view. Asterisks (*) denote $P < 0.05$.

| | F | post-hoc (P) | | | r | | | ICC$_{C-1}$ | | | ICC$_{A-1}$ | | |
|---|---|---|---|---|---|---|---|---|---|---|---|---|---|
| | | MC v C$_L$ | MC v C$_R$ | C$_L$ v C$_R$ | MC v C$_L$ | MC v C$_R$ | C$_L$ v C$_R$ | MC v C$_L$ | MC v C$_R$ | C$_L$ v C$_R$ | MC v C$_L$ | MC v C$_R$ | C$_L$ v C$_R$ |
| Step Time | 0.7 | - | - | - | 0.998* | 0.997* | 0.995* | 0.998* | 0.997* | 0.995* | 0.998* | 0.997* | 0.995* |
| Stance Time | 9.8* | 0.002 | 0.001 | 1.000 | 0.991* | 0.992* | 0.991* | 0.991* | 0.992* | 0.991* | 0.987* | 0.988* | 0.991* |
| Swing Time | 12.5* | <0.001 | 0.001 | 1.000 | 0.963* | 0.958* | 0.964* | 0.963* | 0.955* | 0.962* | 0.941* | 0.936* | 0.963* |
| Double Support Time | 9.2* | 0.002 | 0.003 | 1.000 | 0.909* | 0.898* | 0.894* | 0.909* | 0.897* | 0.893* | 0.873* | 0.861* | 0.896* |
| Step Length | 3.4* | 0.102 | 1.000 | 0.202 | 0.949* | 0.973* | 0.932* | 0.947* | 0.970* | 0.932* | 0.941* | 0.971* | 0.927* |
| Gait Speed | 2.4 | - | - | - | 0.975* | 0.986* | 0.966* | 0.974* | 0.985* | 0.966* | 0.972* | 0.986* | 0.964* |

OpenPose left and right views were 3.1° for the hip, 3.8° for the knee and 5.5° for the ankle. Cross-correlation coefficients showed that the time-varying profiles of hip, knee and ankle angles were well preserved when comparing between OpenPose views (coefficients between 0.880 and 0.992).

When joint angles were assessed along different portions of the walkway, we found that hip and ankle angles had highest values of mean absolute error at the end and beginning of the walkway and lowest values toward the center of the walkway (Fig 8). This suggests that the perspective of the participant relative to the camera affected hip and ankle angle accuracy when calculated with OpenPose. Mean absolute errors in knee angles appeared to be invariant to the position of the participant which suggests that OpenPose knee angles were less sensitive to changes in perspective along the walkway.

## Discussion

As interest in video-based pose estimation of humans [2–7] and animals [8,9] increases, there is considerable potential for using these approaches for fast, inexpensive, markerless gait measurements that can be done in the home or clinic with minimal technological requirement. Pose estimation has been used extensively in human gait classification and recognition [12,18–21]. Here, we provide evidence that pose estimation also shows promise for quantitative spatiotemporal and kinematic analyses of gait that are commonplace in clinical and biomechanical assessments of human walking.

In this study, we aimed to 1) understand how well video-based pose estimation (using OpenPose) could estimate human gait parameters, and 2) provide a workflow for performing human gait analysis with OpenPose. We assessed the accuracy of our OpenPose gait analysis approach by comparing the estimates of spatiotemporal and kinematic gait parameters to measurements obtained by three-dimensional motion capture. We also compared gait parameters estimated by OpenPose analyses from different camera views. We provide a workflow (https://github.com/janstenum/GaitAnalysis-PoseEstimation) for potential users to implement in their own data collection settings and determine whether this approach provides data that are satisfactorily accurate for their research or clinical needs. We also include our interpretations below.

The accuracy of all temporal gait parameters (i.e., step time, stance time, swing time and double support time) was dependent on how well timings of gait events were detected. As a result, the group mean absolute differences of all temporal parameters between motion capture and OpenPose were similar: 0.02 s when comparing individual steps or 0.01 s when comparing individual participant means. The minimal detectable change in temporal gait parameters obtained using three-dimensional motion capture in inter-session, test-retest experiments of healthy human gait have been reported to range from 0.02 to 0.08 s [22,23]. The mean absolute

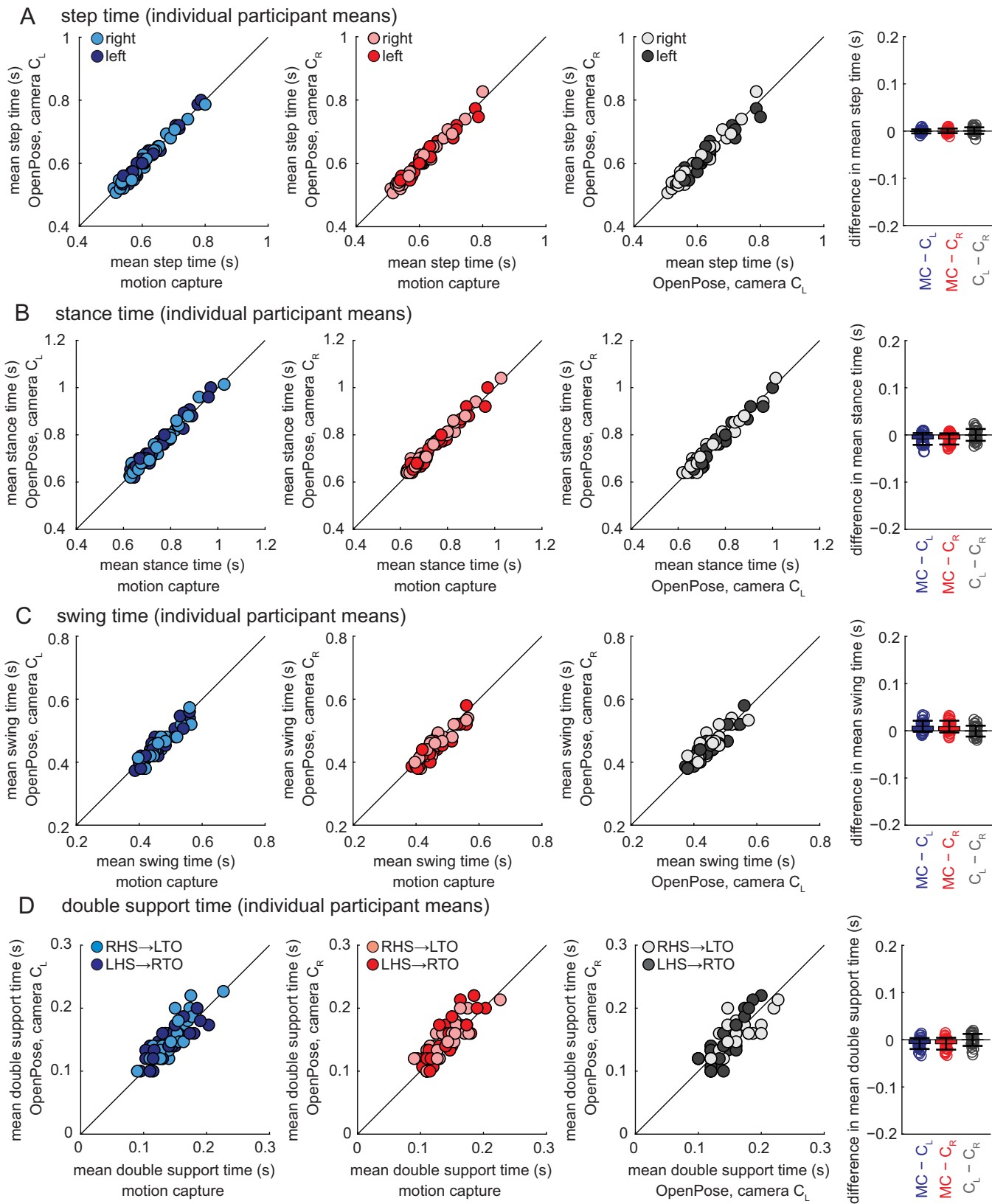

**Fig 3. Temporal gait parameters calculated as individual participant means shown for all participants and measurement systems.** Panels show A) step time, B) stance time, C) swing time, and D) double support time. Bar plots on the far right show individual data, group means, and SD to visualize the distribution of the mean differences observed between the measurement systems. Results from statistical analyses are shown in Tables 4 and 5.

errors in temporal parameters obtained with OpenPose in the current study are less than the differences that may arise from natural variation in the walking pattern between repeated testing sessions. This suggests that OpenPose could detect changes in temporal gait parameters of

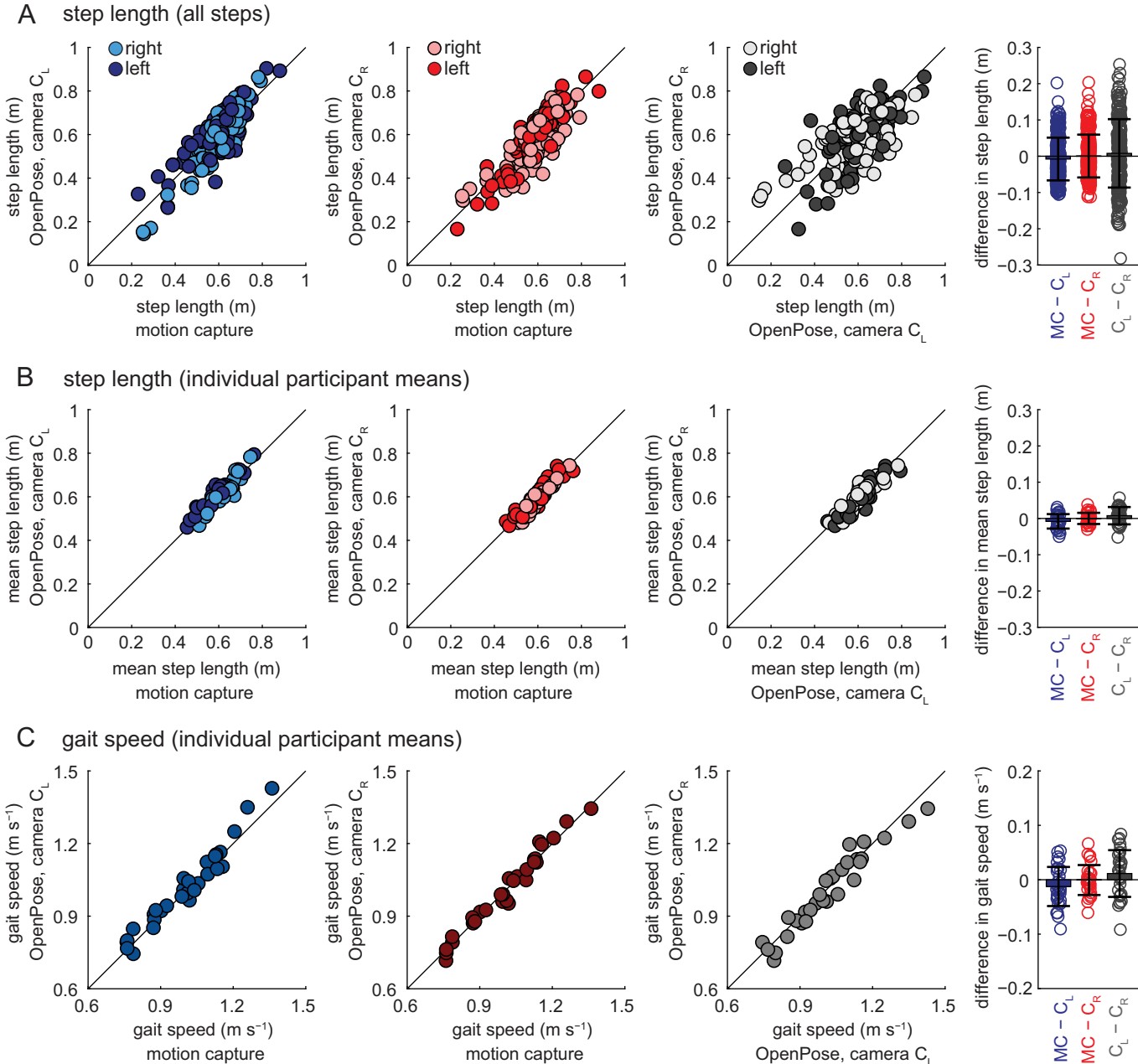

**Fig 4. Step length and gait speed comparisons among the different measurement systems.** A) Step lengths calculated for all individual steps for all participants and measurement systems. B) Step lengths calculated as individual participant means for all participants and measurement systems. C) Gait speeds calculated as individual participant means for all participants and measurement systems. Bar plots on the far right show individual data, group means, and SD to visualize the distributions of the differences observed between the measurement systems. Results from statistical analyses are shown in Tables 2–5.

### differences in step length along the walkway

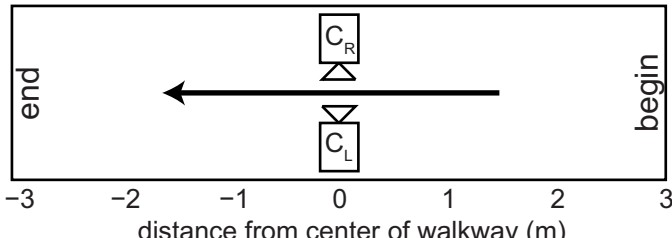

**A** motion capture vs. OpenPose, camera $C_L$

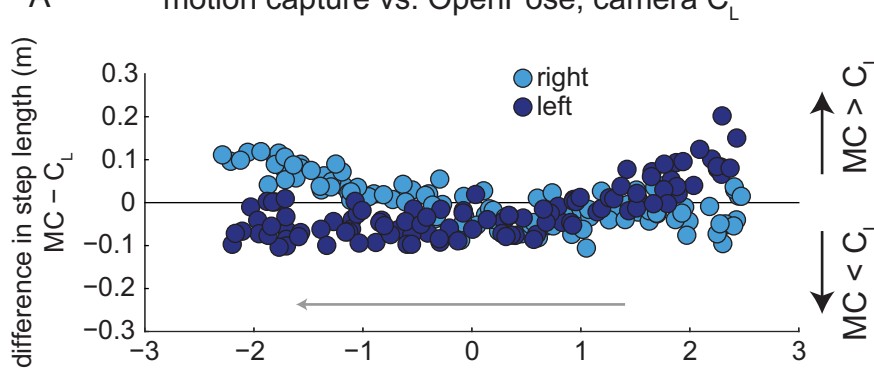

**B** motion capture vs. OpenPose, camera $C_R$

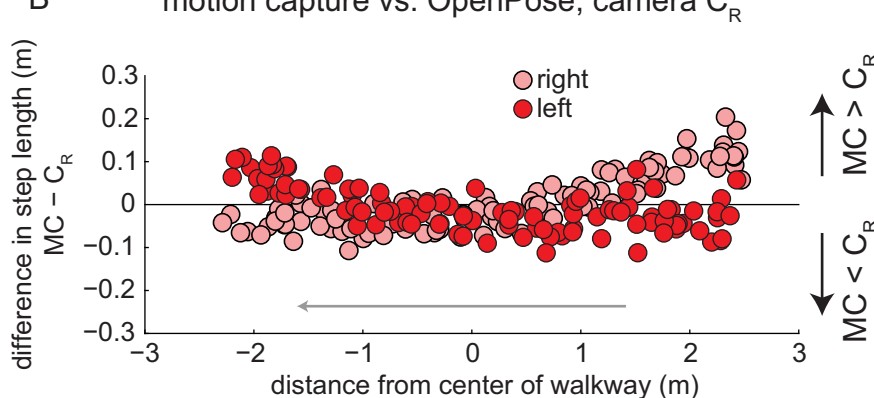

**C** OpenPose, camera $C_L$ vs. OpenPose, camera $C_R$

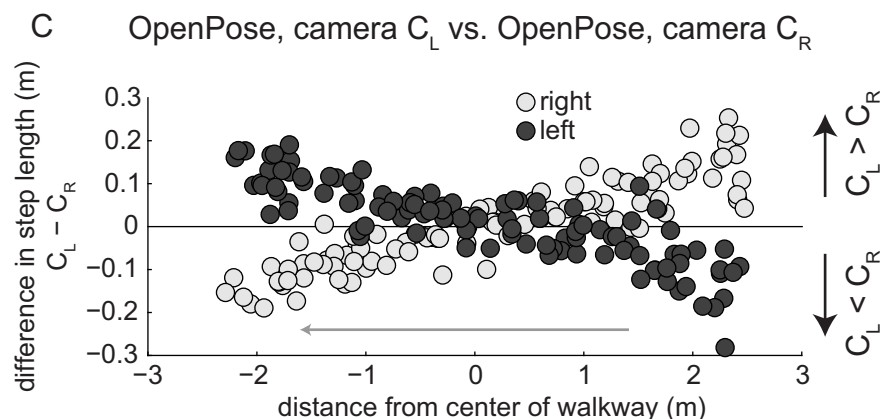

**Fig 5. Differences in step lengths calculated from each measurement system in relation to anterior-posterior position on the walkway.** Dark circles represent left step lengths, light circles represent right step lengths. A) Differences in step lengths calculated by motion capture and OpenPose left ($C_L$) side views across the walkway. B) Differences in step lengths calculated by motion capture and OpenPose right ($C_R$) side views across the walkway. C) Differences in step lengths calculated by OpenPose left ($C_L$) and right ($C_R$) side views across the walkway.

healthy human gait. Note that the short duration of a temporal parameter such as double support time (mean value of 0.14 s in this study) may have influenced that OpenPose estimated its value with relatively poorer precision considering that the temporal resolution of video recordings was 0.04 s in this study.

From the analysis of differences in step length and anterior-posterior position of the participant along the walkway (Figs 5 and 6), we identify the following considerations that affect the estimation of individual step lengths with OpenPose: 1) step length estimation is influenced by the position of the participant along the field of view of the camera, 2) step length estimation is influenced by whether step length is measured at left or right heel-strike, and 3) individual step lengths are estimated most accurately when the person is in the center of the field of view of the camera. If only averaged step lengths are of interest for each participant—perhaps, for example, as a summary statistic in a clinical assessment—OpenPose estimated step length with higher precision in the current study because the systematic errors in step length that occur due to position along the walkway appear to offset when step lengths are averaged across the entire walking bout. Note that averaging step length across the walking bout may not increase accuracy in other scenarios that are different from the current study: e.g., if the participant walks on a treadmill that is not centered in the field of view of the camera, or if the participant walks along an overground walkway but the camera is not centered along the section of the path where the participant is visible. When comparing individual participant mean values of step length, the group mean absolute difference between motion capture and OpenPose was less than 0.020 m and the greatest difference was less than 0.060 m. This suggests that Open-Pose is capable of step length estimation with the accuracy to detect changes in healthy gait, as these values are less than the minimal detectable change in step length of 0.060 m that have been reported in inter-session, test-retest experiments of healthy human gait [22,23].

Gait speed was calculated by dividing step length with step time as the averaged speed across the walking bout. Group mean absolute differences in gait speed between motion capture and OpenPose was 0.03 m s$^{-1}$ and the greatest difference was 0.09 m s$^{-1}$. These values are less than reported test-retest minimal detectable change in gait speed of healthy humans [22,23] which suggest that gait speed can be accurately assessed by OpenPose.

Mean absolute errors in sagittal plane hip, knee and ankle angles were 4.0˚, 5.6˚ and 7.4˚, respectively. Test-retest errors of sagittal plane lower-limb joint angles have previously been reported to be less than 4˚ [24]. Minimal detectable change in sagittal plane peak flexion or extension angles have been reported in the range of 4˚ to 6˚ for hip and knee angles and about 4˚ for ankle ankles [25]. Overall, this suggests that hip and knee angles derived with OpenPose can detect true changes to the gait pattern across test sessions while OpenPose does not possess the precision to confidently detect small changes in ankle angle.

Our OpenPose workflow relies on several post-processing steps, some of which were completed manually to clean the data. In about 20% of the video frames analyzed, multiple persons were detected by OpenPose—as only one person was visible in the videos, the additionally detected persons were false positive identifications by OpenPose (note: there is an OpenPose flag available for limiting the maximum number of persons that can be detected in a given frame, should the user be interested in applying it). In this analysis, we manually checked that the person tracked by OpenPose was the participant. Person detection may be an important

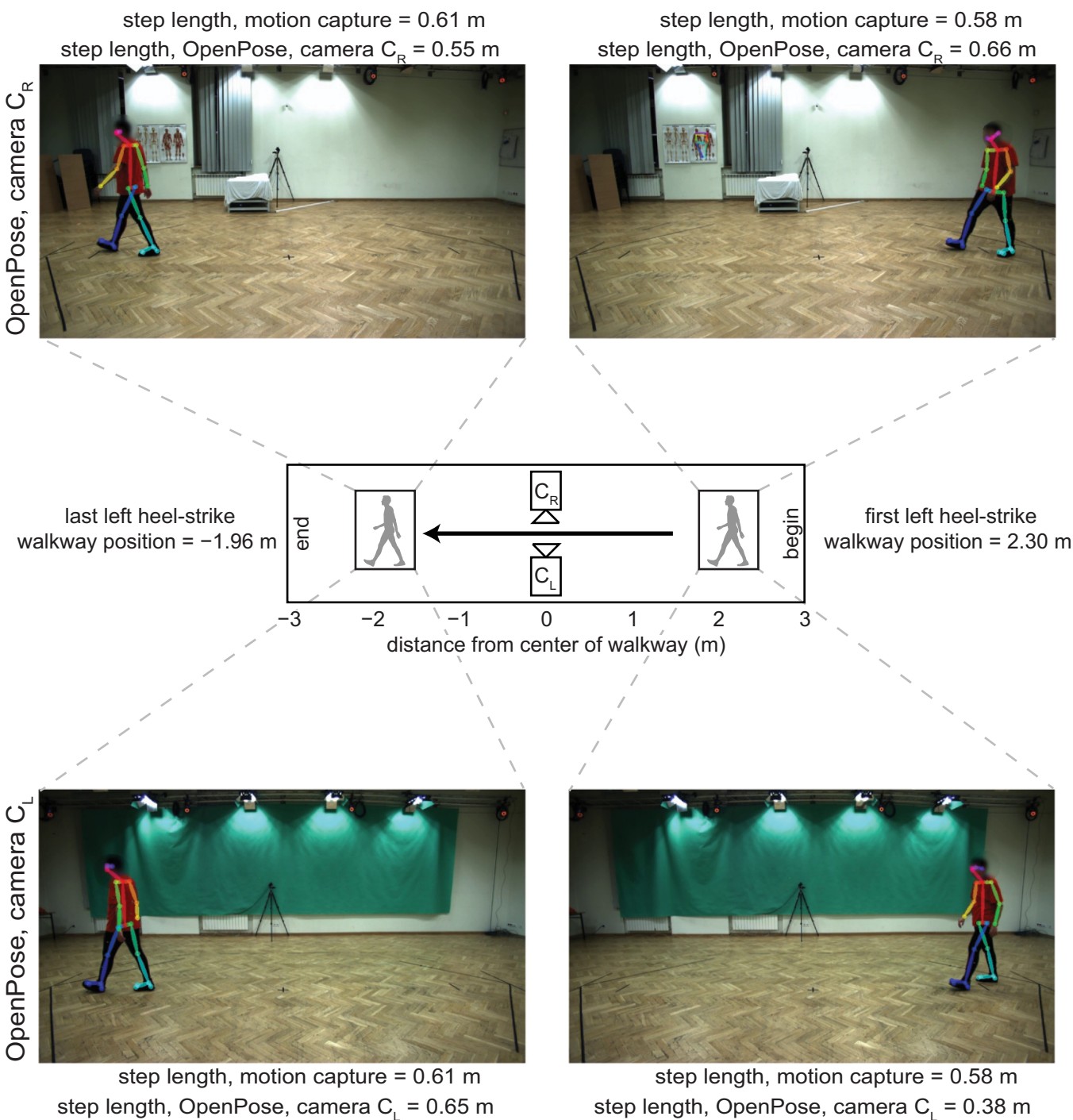

**Fig 6. Example image frames of step length errors.** Representative image frames taken from OpenPose output videos highlighting the discrepancies in step length calculation at different positions along the walkway and from different camera views. Note that images from camera $C_R$ are reflected across a vertical axis to make the walking direction consistent across camera views. The public dataset [18] from which the images belong is made available at http://bytom.pja.edu.pl/projekty/hm-gpjatk/. See Release Agreement for copyrights and permissions.

issue if OpenPose is used to analyze videos in which several individuals are present (e.g., pathological gait in which additional persons are present for safety reasons). We also note that left-right limb identification was switched in about 5% of the analyzed video frames and we

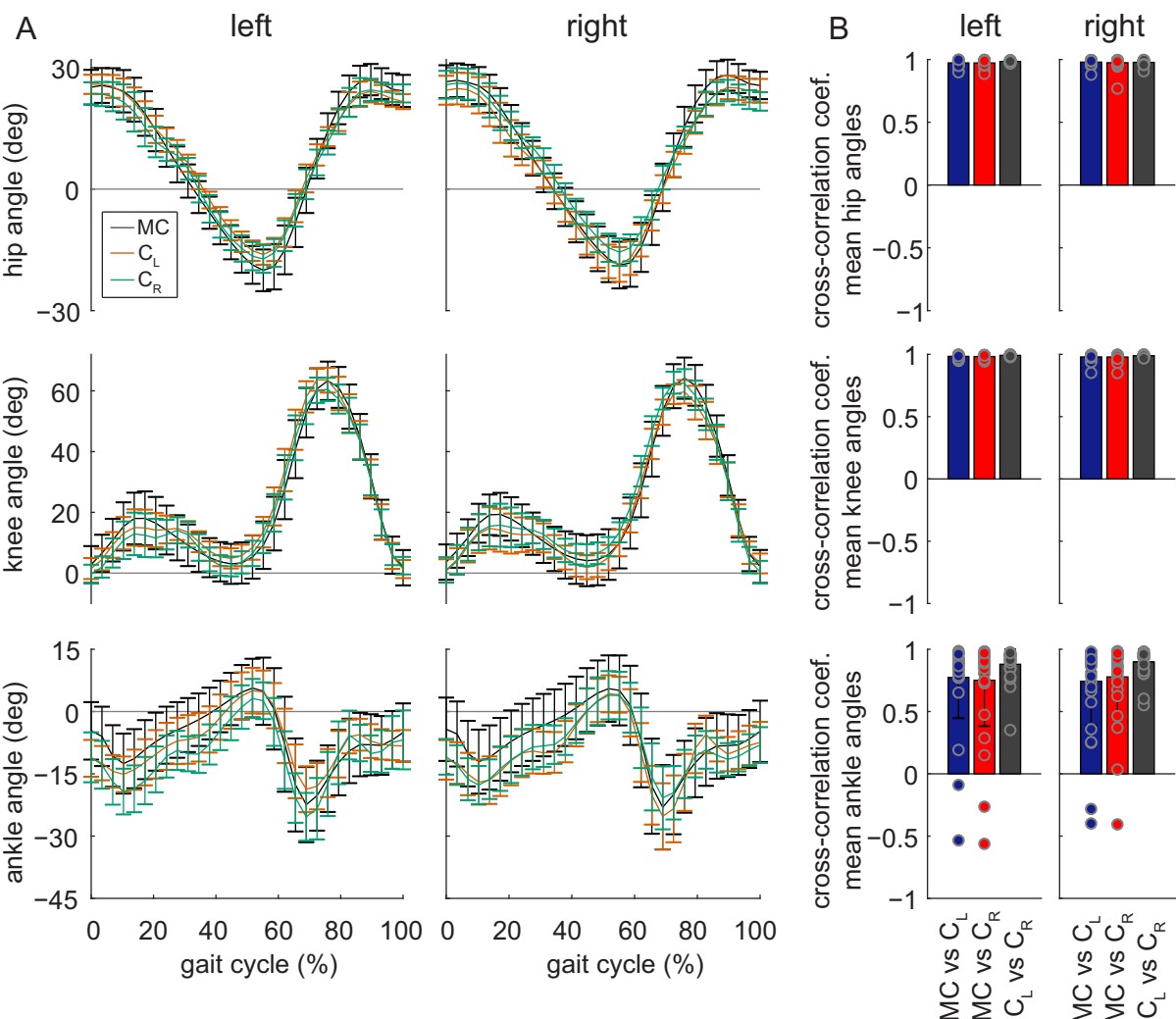

**Fig 7. Sagittal plane hip, knee and ankle angles.** A) Group mean ± SD ensemble sagittal left and right joint angles for the three measurement systems: motion capture (MC), OpenPose left-side view ($C_L$) and right-side view ($C_R$). For all angles, positive values indicate flexion (or dorsiflexion) and negative values indicate extension (or plantarflexion). B) Cross-correlation coefficients at time lag zero (individual data, group means and SD) for individual participant mean joint angle profiles between measurement systems.

**Table 6. Mean absolute error comparisons of joint angles.** *MC*: motion capture; $C_L$: OpenPose left-side view; $C_R$: OpenPose right-side view.

|  |  | N | mean±SD | | |
|---|---|---|---|---|---|
|  |  |  | *MC−$C_L$* | *MC−$C_R$* | *$C_L$−$C_R$* |
| hip (°) | left | 31 | 3.8±1.6 | 4.0±2.1 | 2.6±0.6 |
|  | right | 31 | 3.7±2.0 | 4.0±2.2 | 3.1±1.3 |
| knee (°) | left | 31 | 5.1±2.1 | 5.5±2.4 | 3.5±1.1 |
|  | right | 31 | 5.6±2.7 | 5.6±2.9 | 3.8±1.4 |
| ankle (°) | left | 31 | 6.3±3.4 | 7.4±4.8 | 5.5±2.2 |
|  | right | 31 | 7.4±4.6 | 6.4±3.8 | 4.8±1.8 |

**Table 7. Cross-correlations of joint angles.** *MC*: motion capture; $C_L$: OpenPose left-side view; $C_R$: OpenPose right-side view.

| | | N | mean±SD | | |
|---|---|---|---|---|---|
| | | | $MC \, v \, C_L$ | $MC \, v \, C_R$ | $C_L \, v \, C_R$ |
| hip | left | 31 | 0.973±0.024 | 0.972±0.027 | 0.985±0.007 |
| | right | 31 | 0.979±0.022 | 0.974±0.040 | 0.977±0.018 |
| knee | left | 31 | 0.984±0.012 | 0.983±0.012 | 0.992±0.005 |
| | right | 31 | 0.980±0.027 | 0.979±0.027 | 0.989±0.008 |
| ankle | left | 31 | 0.774±0.327 | 0.751±0.368 | 0.880±0.119 |
| | right | 31 | 0.743±0.354 | 0.778±0.303 | 0.898±0.100 |

manually corrected instances of switched limb identification. The manual portions of the workflow were all based on simple visual inspection of parts of the data and we expect that users with only a basic knowledge of human motion will be able to complete the workflow.

To apply the OpenPose analysis described in the current study, it is important to consider the choice of data collection setup. We used a dataset that contained stationary video camera recordings of sagittal plane views of healthy human gait. It is possible that factors such as camera height and the distance to plane of progression may influence the results. Furthermore, camera angles that deviate from perpendicular will likely affect results. We expect that temporal parameters are most robust to changes in the data collection setup as it is likely that gait events can be reliably detected with a variety of conditions; on the other hand, step lengths and joint angles are likely more susceptible to be affected by the setup condition (e.g., perspective changes from different camera angles).

The results of the current study were based on healthy human gait—does pathological gait affect how well gait parameters and kinematics are estimated with the described workflow? As an example, factors such as body and limb postures may affect how OpenPose tracks keypoints. If keypoints are tracked differently in pathological gait that may potentially have subsequent effects on our workflow and the way in which gait parameters are calculated. We suggest that separate analyses of a range of gait pathologies are needed to establish the accuracy obtained with our workflow.

Some prior studies have used OpenPose to investigate particular features of walking or other human movement patterns [11,12,14,15,26–28]. Our findings align with these reports in that we found OpenPose to show promise in providing quantitative information about human movement (in our case, walking). We also showed that OpenPose estimates of individual participant's mean values of human gait parameters are similar across different camera views, an important confirmation since occlusion is often a primary concern when performing two-dimensional movement analysis. Even though averaged step lengths across the walking bout were similar between OpenPose views (mean absolute difference of 0.020 m), we note that step lengths for individual steps differed by up to 0.282 m between OpenPose views because of the different perspectives (Figs 5C and 6). We anticipate that in-home and clinical video-based analyses will be performed on videos taken by smartphone, tablets, or other household electronic devices. Many of these devices have standard frame rates of 30 Hz during video recording (and capabilities of up to 240 Hz during slow-motion video recording). The frame rate and resolution of the videos used in our analysis (25 Hz and 960x540, respectively) were lower than the factory settings of most modern smartphones (30 Hz, 1920x1080), suggesting that accuracy may even be improved when using household devices.

Many other markerless motion capture approaches exist for human gait analysis. These include silhouette analyses [29–32], commercially available products like the Microsoft Kinect

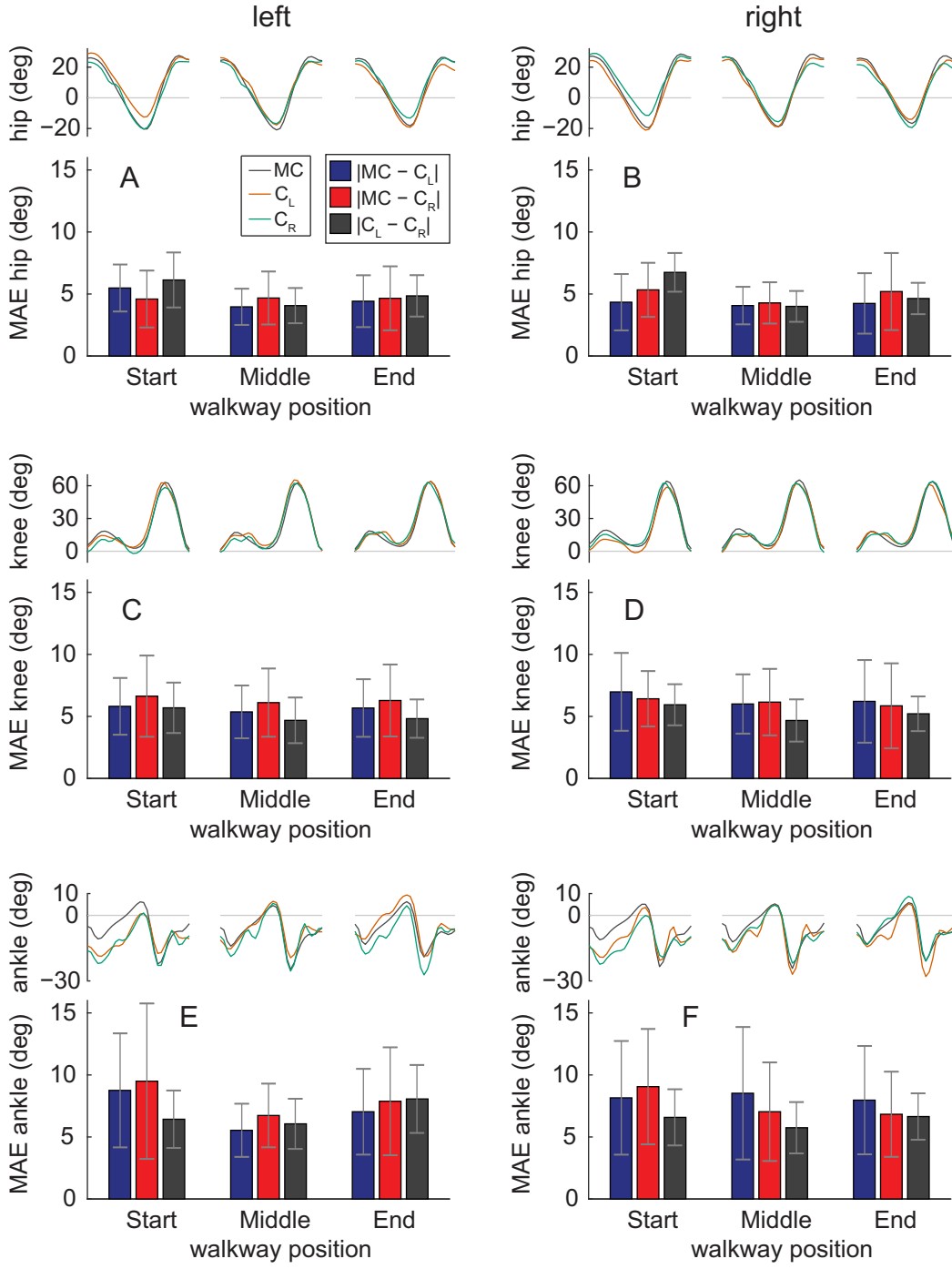

**Fig 8. Effect of walkway position on sagittal plane joint angles.** Comparisons of joint angles and mean absolute error (MAE) between measurement systems at three portions of the walkway. The portions were based on the average anterior-posterior position of the C7 marker in motion capture data throughout each gait cycle. Gait cycles were binned in regions: 'Start' spanning anterior-posterior positions less than −0.50 m of the middle of the walkway, 'Middle spanning position from −0.50 to 0.50 m, and 'End' spanning positions greater than 0.50 m. The number of gait cycles analyzed were 30, 22 and 24 for 'Start', 'Middle' and 'End', respectively, for the left leg and 31, 21 and 26 for the right leg.

[20,33–38], and a variety of other technologies [18,39–42]. We did not directly compare the results of our OpenPose analyses to results of any of these other markerless approaches, and thus we are hesitant to speculate about the relative accuracy of our approach against others. Given the results of several studies cited above, we consider it likely that some of these methods could produce more accurate results. However, a significant advantage of video-based pose estimation with particular relevance for clinic- and home-based gait analysis is that data collection for pose estimation requires no equipment beyond a digital video recording device, whereas many of the other methods require expensive, less accessible, and less portable equipment. We further discuss our observations on the pros and cons of using OpenPose for human gait analysis in the following section.

## Suggestions and limitations

**Video recordings.** Because we did not record the videos used in this study, we did not experiment with different video recording techniques. However, we expect that recording methods with higher frame rates (e.g., the slow-motion video recording feature available on most smartphones), faster shutter speeds, and wider fields of view (and distortion corrected) may improve the accuracy of video-based gait analyses. It is also likely that changes in lighting, clothing and footwear of the participant may affect the ability to track specific keypoints. For example, loose-fitting clothing may introduce more ambiguity into estimations of hip or knee keypoints.

**Recording in the home or clinic.** To generate estimates of gait parameters that incorporate spatial information (e.g., gait speed, step length), it is necessary to scale the video. Here, we accomplished this by scaling the video to known measurements on the ground. This could be done in the home or clinic by placing an object of known size in the field of view at the same depth with which the person is walking or use a known anthropometric quantity such as body height.

The videos used in our study were recorded in a large open room with ample distance between the participant and the camera. This allowed recording of several strides per trial and clear sagittal views of the participant. Given the limited availability of large, open spaces in the home or clinic, we perceive a need for approaches capable of estimating at least spatiotemporal gait parameters using frontal plane recordings that are more amenable to narrow spaces.

OpenPose is also capable of three-dimensional human movement analysis. However, this requires multiple simultaneous camera recordings. Because we assumed that most videos taken in the home or clinic will be recorded by a single device, we limited this study to two-dimensional analyses of human walking.

**Network selection.** Here, we used the demo and pre-trained network provided by Open-Pose because we considered that this is the most accessible approach that is likely to be used by most new users of OpenPose, especially those who do not have significant expertise in engineering or computer science. Another advantage of using the pre-trained network is that it has already been trained on thousands of images, saving the user significant time and effort in training their own network. However, it may be possible to obtain more accurate video-based analyses by training gait-specific networks from different views (e.g., sagittal, frontal). Similarly, if a user aims to study clinical populations with gait dysfunction (e.g., stroke, Parkinson's disease, spinal cord injury) or children, it may be beneficial to train networks that are specific to each population. Furthermore, we did not compare our workflow to other available pose estimation algorithms (e.g., DeeperCut [4], DeepLabCut [8], DeepPose [7], AlphaPose [43]). These approaches are evolving rapidly and are likely to continue to improve in the near future.

**Potential sources of error in video-based estimates of gait parameters vs. motion capture.** We considered several potential reasons for discrepancies between the parameters

estimated by OpenPose and those measured using three-dimensional motion capture. Some sources of error may be intrinsic to OpenPose. First and most obvious, OpenPose does not track movements of the human body perfectly from frame-to-frame. Second, the body key-points identified by OpenPose are unlikely to be equivalent to the marker landmarks. Open-Pose relies on visually labeled generalized keypoints (e.g., "ankle", "knee") whereas marker placement relies on manual palpation of bony landmarks (e.g., lateral malleolus, femoral epicondyle). Certainly, there may also be some degree of error in the placement of the motion capture markers.

Other sources of error may be introduced by the video recording processes. We observed that different perspective toward the edges of the field of view introduced errors into our OpenPose estimates of parameters that rely on spatial information (e.g., step length, Figs 5 and 6). These errors affected estimates of individual steps but appeared to largely offset one another when estimating mean parameters for each participant. Furthermore, blurring of individual frames due to relatively slow shutter speeds and relatively low frame rates may also contribute to inaccuracies in pose estimation. In the dataset that we analyzed here, participants walked at a relatively slow speed of about 1.0 m s$^{-1}$. Tracking faster walking or running could blur images leading to poorer tracking by pose estimation algorithms.

## Conclusions

Here, we observed that pose estimation (using OpenPose) can provide estimates of many human gait parameters at an accuracy required to detect changes to the gait pattern in healthy humans. The pose estimation approach used in this study requires only a two-dimensional digital video input and outputs a wide array of spatiotemporal and kinematic gait parameters. We identified and discussed features of our approach that we believe may have influenced the accuracy of the OpenPose estimation of particular gait parameters, and we provided a workflow that is available at https://github.com/janstenum/GaitAnalysis-PoseEstimation. We are optimistic about the potential that this initial study reveals for measuring human gait data in the sagittal plane using video-based pose estimation and expect that such methods will continue to improve in the near future.

## Material and methods

### Ethics statement

We used a publicly available dataset [18] of overground walking sequences from 32 healthy participants (22 men and 10 women) made available at http://bytom.pja.edu.pl/projekty/hm-gpjatk/. The dataset included synchronized three-dimensional motion capture files and digital video recordings of the walking sequences. The dataset does not contain identifiable participant information and faces have been blurred in the video recordings. Our analyses of these previously published videos were deemed exempt by the Johns Hopkins Institutional Review Board.

### Walking sequences

The laboratory space in which the participants walked included ten motion capture cameras and four video cameras that recorded left- and right-side sagittal plane views and front and back frontal plane views (see Fig 1A for overview of laboratory space). We used a subset of the data (sequences labelled s1) that consisted of a single walking bout of about five meters that included gait initiation and termination. We excluded data for one participant because the data belonged to another subset with diagonal walking sequences. Therefore, we analyzed 31

total gait trials (one trial per participant). The mean ± SD duration of the video recordings was 5.12 ± 0.73 s.

## Data collection

The motion capture cameras (Vicon MX-T40, Denver, CO, USA) recorded three-dimensional marker positions at 100 Hz. Motion was recorded by tracking markers that were placed on the seventh cervical vertebrae (C7), tenth thoracic vertebrae (T10), manubrium, sternum, right upper back and bilaterally on the front and back of the head, shoulder, upper arm, elbow, forearm, wrist (at radius and ulna), middle finger, anterior superior iliac spine (ASIS), posterior superior iliac spine (PSIS), thigh, knee, shank, ankle, heel and toe.

Four video cameras (Basler Pilot piA1900-32gc, Ahrensburg, Germany) recorded left (camera $C_L$) and right (camera $C_R$) side sagittal plane views and front and back frontal plane views of the walking sequences at 25 Hz. We only analyzed video recordings of left and right sagittal plane views. The digital camera images were RBG files with 960x540 pixel resolution. Motion capture and video recording were synchronized so that the time of every fourth motion capture data point corresponded to each time point of the video frames. Cameras were mounted on tripods and the height was set about 1.3 m. The distance from cameras $C_L$ or $C_R$ to the participants was about 3.3 m.

## Data processing

Motion capture data had already been smoothed and were therefore not processed further. We used the following workflow to process sagittal plane video recordings and obtain two-dimensional coordinates of OpenPose keypoints. We first analyzed the video recordings with Open-Pose using our provided Google Colaboratory notebook and next processed the data using custom written MATLAB software (also provided). The workflow is shown in Fig 1B and is as follows (detailed instructions, sample videos, and software for all steps can be found at https://github.com/janstenum/GaitAnalysis-PoseEstimation):

1. OpenPose analyses

   a. We used Google Colaboratory to run OpenPose analyses of the sagittal plane video recordings. Google Colaboratory executes Python code and allows the user to access GPUs remotely through Google cloud services. This allows for much faster analysis than can be executed on a CPU. Note that it was possible to use Google Colaboratory for our study because we analyzed publicly available videos; users that aim to analyze videos of research participants or patients may need to run the software locally through a Python environment to avoid uploading identifiable participant or patient information into Google Drive if this is not deemed sufficiently secure by the user's institution. In our Google Colaboratory notebook, we also provide code for analyzing YouTube videos with OpenPose should this be of interest to some users.

   b. Video recordings were analyzed in OpenPose using the BODY_25 keypoint model that tracks the following 25 keypoints: nose, neck, midhip and bilateral keypoints at eyes, ears, shoulders, elbows, wrists, hips, knees, ankles, heels, big toes and small toes.

   c. The outputs of the OpenPose analyses yielded: 1) JSON files for every video frame containing pixel coordinates (origin at upper left corner of the video) of each keypoint detected in the frame, and 2) a new video file in which a stick figure that represents the detected keypoints is overlaid onto the original video recording. The JSON files were then downloaded for further offline analysis in MATLAB.

2. MATLAB processing steps

 a. In about 20% of the total number of frames (3,970), OpenPose detected false positive persons (e.g., from an anatomy poster that was visible in some videos or from tripods that were visible). We visually inspected all frames in which multiple persons were detected so that the keypoints tracked were always the keypoints from the participant. OpenPose consistently assigned the participant as person ID #1 in all frames wherein multiple persons were detected. We have since updated our Google Colaboratory notebook to limit OpenPose to identification of one person per video. As a result, we have not provided this step in our workflow.

 b. We changed the pixel coordinate system so that positive vertical was directed upward and positive horizontal was in the direction of travel. The location of the origin depended on left (camera $C_L$) or right (camera $C_R$) side views of the video recording. For the left-side view, the origin was set at the lower right corner; for the right-side view, the origin was set at the lower left corner. Note that all our further analyses were invariant to the location of the origin.

 c. In approximately 5% of the total frames, OpenPose erroneously switched the left-right identification of the limbs. We visually inspected anterior-posterior trajectories of the left and right ankle to identify and correct the keypoints hip, knee, ankle, heel, big toe and small toe in these frames. We also corrected frames in which keypoints on the left and right legs were identified on the same leg.

 d. We filled gaps in keypoint trajectories (i.e., frames where OpenPose did not detect all keypoints) using linear interpolation for gaps spanning up to 0.12 s (i.e., for gaps spanning up to two video frames).

 e. We filtered trajectories using a zero-lag 4$^{\text{th}}$ order low-pass Butterworth filter with a cut-off frequency at 5 Hz.

 f. Last, we calculated a scaling factor to obtain dimensionalized coordinate values from the pixel coordinates. The scaling factor ($s$) was calculated as:

$$s = \frac{distance}{pixel\ length}.$$

We used the distance between two strips of tapes on the floor at each end of the walkway that ran parallel to the viewpoints of cameras $C_L$ and $C_R$ (see Fig 1A). Pixel length was taken as the horizontal pixel length between the midpoints of each strip of tape. We then calculated dimensionalized coordinates as:

$$x_{dim} = s * x_{pixel},$$

and

$$y_{dim} = s * y_{pixel}.$$

The distance between the strips of tape was not measured during the original data collection for the dataset and the tape has since been removed from the floor. We therefore estimated the distance between the strips of tape by: 1) calculating individual scaling factors from each participant based on the horizontal distance traversed by left and right ankle markers in motion capture data and left and right ankle keypoints in OpenPose data; 2) calculating the distance between the strips of tape for each individual participant; 3) calculating the ensemble mean

distance. The ensemble mean distance was 6.30 m and we used this fixed value to calculate scaling factors for each individual participant.

To examine how robustly the known distance was estimated, we also calculated the distance between the strips of tape by using the 'CLAV' marker (placed on the manubrium) from the motion capture dataset and the neck marker in OpenPose and obtained an ensemble mean distance of 5.91 m (2.5% reduction relative to the distance of 6.30 m that we used here). This means that there is a margin of uncertainty associated with the scaling factor that we used to dimensionalize the pixel coordinates obtained with OpenPose.

Note that the use of the scaling factor assumes that the participants walked perpendicularly and at a fixed depth relative to the cameras. Barring natural side-to-side fluctuations in gait, we find that this is a reasonable assumption since net displacement of medio-lateral position of the C7 marker in the motion capture data from the start to the end of the walking sequence was (ensemble mean ± SD) 0.069 ± 0.047 m.

We calculated event timings of left and right heel-strikes and toe-offs in motion capture data and data of OpenPose left ($C_L$) and right ($C_R$) side views by independently applying the same method to each set of data [44]. Heel-strikes and toe-offs were defined by the time points of positive and negative peaks of the anterior-posterior ankle trajectories relative to the pelvis (midpoint of left and right ASIS and PSIS markers in motion capture data; midhip keypoint in OpenPose data). All processing steps were completed by one researcher (JS).

We calculated the following spatiotemporal gait parameters in the motion capture and OpenPose data:

- Step time: duration in seconds between consecutive bilateral heel-strikes.

- Stance time: duration in seconds between heel-strike and toe-off of the same leg.

- Swing time: duration in seconds between toe-off and heel-strike of the same leg.

- Double support time: duration in seconds between heel-strike of one leg and toe-off of the contralateral leg.

- Step length: anterior-posterior distance in meters between left and right ankle markers (motion capture) or ankle keypoints (OpenPose) at heel-strike.

- Gait speed: step length divided by step time.

For step time, right step refers to the duration until right heel-strike and vice versa for the left step. For step length, right step refers to the distance between the ankles at right heel-strike and vice versa for the left step. We calculated step time, stance time, swing time, double support time and step length for all steps and as averages for individual participants. Gait speed was calculated from individual participant means of step time and step length.

We calculated sagittal plane hip, knee and ankle angles of left and right legs using two-dimensional marker (motion capture) and keypoint (OpenPose) coordinates. The hip joint center in the motion capture data was estimated based on a regression model [45]. We used the following markers (motion capture) or keypoints (OpenPose) to calculate joint angles: hip angle was the vector between hip and knee (0° is vertical, flexion is positive, extension is negative); knee angle was the angle formed by the vectors between hip and knee and between knee and ankle (0° is vertical, flexion is positive, extension is negative); ankle angle was the angle formed by the vectors between knee and ankle and between ankle and toe marker or big toe keypoint (for motion capture and OpenPose, respectively) (0° is horizontal, dorsiflexion is positive, plantarflexion is negative). All joint angles were calculated as the mean across the stride cycle of individual participants. Mean absolute errors across the gait cycle were

calculated for all joint angles between the three measurement systems. We calculated cross-correlations at time lag zero to assess the similarity between mean joint angle trajectories (i.e., one measure per participant) calculated by motion capture and OpenPose left ($C_L$) side view, by motion capture and OpenPose right ($C_R$) side view, and by OpenPose left ($C_L$) and right ($C_R$) side views. We assessed the effect of participant position along the walkway by calculating mean absolute errors of joint angles of individual gait cycles between measurement systems as a function of the average anterior-posterior position of the C7 marker in motion capture data throughout each gait cycle. Gait cycles were binned in regions: 'Start' spanning anterior-posterior positions less than −0.50 m of the middle of the walkway, 'Middle spanning position from −0.50 to 0.50 m, and 'End' spanning positions greater than 0.50 m.

## Statistical analyses

We obtained gait event (i.e., heel-strike and toe-off) times, spatiotemporal gait parameters and sagittal joint angles from three measurement systems: motion capture, OpenPose left ($C_L$) and right ($C_R$) side views. We used one-way repeated measures ANOVA to assess potential differences in gait parameters and peak joint angles among measurement systems. In the event of a statistically significant main effect, we performed post-hoc pairwise comparisons with Bonferroni corrections. We calculated Pearson correlation coefficients ($r$) and intra-class correlation coefficients ($ICC_{C-1}$ and $ICC_{A-1}$) of spatiotemporal gait parameters to assess correlations ($r$), consistency ($ICC_{C-1}$) and agreement ($ICC_{A-1}$) between 1) motion capture and OpenPose left ($C_L$) side view, 2) motion capture and OpenPose right ($C_R$) side view and 3) OpenPose left ($C_L$) and right ($C_R$) side views. We set the level of significance at 0.05 for all analyses.

## Acknowledgments

We thank Amy Bastian, Darcy Reisman, and their lab members for helpful comments. We also thank Tomasz Krzeszowski for answering questions about the dataset used in this study.

## Author Contributions

**Conceptualization:** Jan Stenum, Cristina Rossi, Ryan T. Roemmich.

**Data curation:** Jan Stenum.

**Formal analysis:** Jan Stenum.

**Funding acquisition:** Ryan T. Roemmich.

**Investigation:** Jan Stenum, Cristina Rossi, Ryan T. Roemmich.

**Methodology:** Jan Stenum, Cristina Rossi, Ryan T. Roemmich.

**Project administration:** Ryan T. Roemmich.

**Resources:** Jan Stenum, Ryan T. Roemmich.

**Software:** Jan Stenum.

**Supervision:** Ryan T. Roemmich.

**Validation:** Jan Stenum.

**Visualization:** Jan Stenum, Ryan T. Roemmich.

**Writing – original draft:** Jan Stenum, Ryan T. Roemmich.

**Writing – review & editing:** Jan Stenum, Cristina Rossi, Ryan T. Roemmich.

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
