## [Decision Letter · Decision Letter 0]

10 Dec 2020

Dear Dr. Stenum,

Thank you very much for submitting your manuscript "Two-dimensional video-based analysis of human gait using pose estimation" for consideration at PLOS Computational Biology.

As with all papers reviewed by the journal, your manuscript was reviewed by members of the editorial board and by several independent reviewers. In light of the reviews (below this email), we would like to invite the resubmission of a significantly-revised version that takes into account the reviewers' comments.

The reviewers raised several important points. Specifically, please address the applicability of the method. How likely/ready the method is for a clinical application? Is it just proof of concept? In addition, the reviewers are concerned regarding the accuracy measures. Is it possible to quantify the accuracy?

We cannot make any decision about publication until we have seen the revised manuscript and your response to the reviewers' comments. Your revised manuscript is also likely to be sent to reviewers for further evaluation.

Sincerely,

Dina Schneidman

Software Editor

PLOS Computational Biology

Reviewer's Responses to Questions

**Comments to the Authors:**

Reviewer #1: Major Comments:

This study investigates the accuracy of vision-based motion analysis in enabling extraction of biomechanically and clinically relevant outcomes, such as spatiotemporal and kinematic parameters of gait. Given the increasing popularity of vision-based techniques, and their potential to enable deployment of lightweight gait analysis tools across clinics and patient homes, this study is relevant and timely. However, the manuscript is crippled by weaknesses in presentation, technical rigor, and interpretation.

The major contribution of the work needs to be clarified: does the work offer a technical or an experimental contribution, or both? The authors seem to suggest both, but do not make a strong case for either. For the experimental component: what were the hypotheses? If not hypothesis-driven (more discovery-driven), could it be stated up front what the authors expected to find and why? For the technical contribution: it is stated that a framework that is built on top of a computer vision algorithm (OpenPose) will enable individuals with no programming background (e.g., clinicians) to use OpenPose. This framework is still provided in Matlab to my understanding and is not generalizable to other datasets without further programing. Will the provided tool generalize to other data collection configurations as is? If the contribution is to enable non-programmers, then further experiments on the generalizability of the provided tools need to be provided. How does the tool perform on data collected elsewhere?

Overall the manuscript provides no interpretation of whether the accuracy of the extracted parameters is sufficient for specific applications in clinical biomechanics. The language is descriptive and imprecise, referring to results as “reasonably accurate,” for example. What may be reasonable to one, might not be to another – scientist, clinician, field. Also, differences of 0.001±0.023 seconds are reported, but this is below the time resolution with which the data were captured.

The manuscript would benefit from further editorial work and restructuring to improve clarity.

Minor Comments:

Line 23: “immobile” in what sense?

Lines 32-36: “well”, “less accurate”, and “improved agreement” are qualitative descriptions and the abstract should include quantitative findings backed up by statistics (no need to include statistical tests in the abstract)

Line 61: “use of a large dataset containing walking sequences of many individuals”

What does “large” mean in this context?

Line 66: “an “out of the box” approach with no need for additional network processing.”

What kind of “out of the box” approach?

Line 73: 135 keypoints? Could you please clarify what these 135 keypoints correspond to? Joint centers across different people in the same image?

Line 87 and Line 92: What is reasonably accurate? I suggest refraining from using subjective verbiage in a scientific article

Lines 94-100: These are methods, not results

Line 104: Please refer to tables parenthetically

Line 109 -111: This is not a result

Lines 118 – 122: These descriptions belong in the Methods section

Line 132: I suggest leaving “somewhat” out and finding more precise language to describe your findings. Overall, these casual terms can diminish the seriousness with which your science is perceived.

Lines 159 – 161: “Pearson and intra-class correlation coefficients of step length between OpenPose left (C1) and right (C3) side views were 0.671 for all steps and ≥0.927 for individual participant means.” Statistics are used to back up a finding, and I recommend that they do not become the topic. It is best to state your finding and back it up parenthetically with the appropriate statistic.

Line 164: “All correlations were statistically significant.” See above.

Lines 174 – 186: Are these findings specific to the video data you used or can anything more generalizable be learned from them?

Lines 187-195: These are not results, but rather discussion points. Also, it seems quite obvious that (1) the quality of OpenPose’s estimation will degrade as the subject moves away from the camera, (2) “step length estimation is influenced by whether step length is measured at left or right heel-strike,” as this relates to the position of the leg versus the camera, and (3) “individual step lengths are estimated most accurately when the person is in the center of the field of view of the camera.” Did the authors not expect these findings?

Line 242: “Gait events (e.g., heel-strikes and toe-offs) identified by OpenPose matched those identified by motion capture very well (i.e., generally within one motion capture frame).” Can you comment on how this is possible when the video data has much lower temporal resolution?

Lines 490 – 502: Calculation of joint angles is unclear.

Reviewer #2: The detailed review is embedded as comments in the PDF.

Overall I found this to be an nteresting paper and very well written. The authors have done a nice job exploring a relevant topic and (for the most part) clearly and transparently presenting their results. The comments embedded in the pdf are mostly minor - though there are quite a few of them.

My one major concern is the authors' choice of correlation as a measure of performance in the comparison to kinematics. In fact, the performance in this realm appears quite poor, contrary to the way the authors portray it. Note that this does NOT make the research less valuable - though it does raise doubts about the authors' devotion to an "out-of-the box" solution.

My impression is that the authors are a bit overly-optimistic about the technology and its potential application to real world (especially clinical) problems. I urge them to temper this enthusiasm with a bit more realism regarding the relatively poor performance of the method on kinematics.

**Have all data underlying the figures and results presented in the manuscript been provided?**

Reviewer #1: None

Reviewer #2: Yes

PLOS authors have the option to publish the peer review history of their article (what does this mean?). If published, this will include your full peer review and any attached files.

Reviewer #1: No

Reviewer #2: No
---

## [Decision Letter · Decision Letter 1]

1 Apr 2021

Dear Dr. Stenum,

We are pleased to inform you that your manuscript 'Two-dimensional video-based analysis of human gait using pose estimation' has been provisionally accepted for publication in PLOS Computational Biology.

Best regards,

Dina Schneidman

Software Editor

PLOS Computational Biology

Reviewer's Responses to Questions

**Comments to the Authors:**

Reviewer #2: The authors have done a thorough and commendable job responding to all of my comments. I appreciate their earnest efforts, and feel the paper is significantly stronger as a result.

I feel the paper will be a useful contribution to this new and growing area.

**Have all data underlying the figures and results presented in the manuscript been provided?**

Reviewer #2: None

PLOS authors have the option to publish the peer review history of their article (what does this mean?). If published, this will include your full peer review and any attached files.

Reviewer #2: No

---

## [Editor Report · Acceptance letter]

16 Apr 2021

PCOMPBIOL-D-20-01556R1 

Two-dimensional video-based analysis of human gait using pose estimation

Dear Dr Stenum,

I am pleased to inform you that your manuscript has been formally accepted for publication in PLOS Computational Biology. Your manuscript is now with our production department and you will be notified of the publication date in due course.

With kind regards,

Katalin Szabo
